# Lipid droplet biology and evolution illuminated by the characterization of a novel perilipin in teleost fish

James G Granneman[1]*, Vickie A Kimler[1†], Huamei Zhang[1], Xiangqun Ye[1], Xixia Luo[2,3], John H Postlethwait[4,5], Ryan Thummel[2,3]*

[1]Center for Integrative Metabolic and Endocrine Research, Wayne State University School of Medicine, Detroit, United States; [2]Department of Anatomy and Cell Biology, Wayne State University School of Medicine, Detroit, United States; [3]Department of Ophthalmology, Wayne State University School of Medicine, Detroit, United States; [4]Institute of Neuroscience, University of Oregon, Eugene, United States; [5]Department of Biology, University of Oregon, Eugene, United States

**Abstract** Perilipin (PLIN) proteins constitute an ancient family important in lipid droplet (LD) formation and triglyceride metabolism. We identified an additional *PLIN* clade (*plin6*) that is unique to teleosts and can be traced to the two whole genome duplications that occurred early in vertebrate evolution. Plin6 is highly expressed in skin xanthophores, which mediate red/yellow pigmentation and trafficking, but not in tissues associated with lipid metabolism. Biochemical and immunochemical analyses demonstrate that zebrafish Plin6 protein targets the surface of pigment-containing carotenoid droplets (CD). Protein kinase A (PKA) activation, which mediates CD dispersion in xanthophores, phosphorylates Plin6 on conserved residues. Knockout of *plin6* in zebrafish severely impairs the ability of CD to concentrate carotenoids and prevents tight clustering of CD within carotenoid bodies. Ultrastructural and functional analyses indicate that LD and CD are homologous structures, and that Plin6 was functionalized early in vertebrate evolution for concentrating and trafficking pigment.

*For correspondence: jgranne@ med.wayne.edu (JGG); rthummel@med.wayne.edu (RT)

**Present address:** [†]Eye Research Institute, Oakland University, Rochester, United States

**Competing interests:** The authors declare that no competing interests exist.

## Introduction

Intracellular structures that assimilate and traffic lipophilic cargo can be found in virtually every eukaryotic cell, including those from algae, yeast, plants, insects, and vertebrates (*Zweytick et al., 2000*; *Arrese et al., 2014*). Perhaps best known of these structures are the intracellular lipid droplets (LD), which typically store neutral lipids like triglyceride and cholesterol esters, and play diverse roles in metabolism and signaling. These intracellular organelles share a simple structure: a central core of lipophilic cargo surrounded by a phospholipid monolayer that is embedded with critical proteins. Until recently, intracellular LD were considered to be inert lipid 'inclusions'; however, growing evidence indicates that LD are bona fide organelles whose function is controlled in part by an evolutionarily-conserved protein network (*Farese and Walther, 2009*; *Walther and Farese, 2012*).

Recent work has revealed the importance of Perilipin (PLIN) proteins in the generation of LD and in regulating protein trafficking on the LD (*Brasaemle et al., 2004*; *Londos et al., 2005*; *Granneman and Moore, 2008*). *PLIN*s comprise an ancient gene family that plays central roles in neutral lipid metabolism (*Miura et al., 2002*). Mammalian PLIN proteins contain a conserved PAT domain (for Perilipin, Adipophilin, TIP47; now named Plin1, Plin2 and Plin3, respectively) that appears to be important in LD targeting (*Londos et al., 2005*). The human genome has five *PLIN*

genes, designated *PLIN1-5*. PLIN2 and PLIN3 are found in most cells and are thought to be involved in the formation and expansion of LD from the smooth endoplasmic reticulum (SER) (*Wolins et al., 2006*; *Ducharme and Bickel, 2008*). In addition, tissue-specific PLINs have evolved in mammals that confer functional diversity. For example, Plin1 mediates hormone-stimulated lipolysis in fat cells, which supply tissues with energy in the form of fatty acids, whereas Plin5 manages fatty acid storage/oxidation in muscle cells (*Moore et al., 2005*; *Wolins et al., 2006*; *Granneman et al., 2011*; *Sanders et al., 2015*).

Most work addressing PLIN protein function has been performed in mammalian systems, focusing on neutral lipid metabolism, but relatively little has been performed on non-mammalian vertebrates. Our database analysis identified a novel *PLIN* clade (*plin6*) that has existed in vertebrates since before the divergence of the fish and human lineages. Plin6 is highly expressed in skin chromatophores (specifically, in xanthophores) that mediate red/yellow pigmentation. Structural and functional analyses indicate that Plin6 is targeted to intracellular carotenoid droplets, where it plays a role in concentrating and trafficking lipophilic pigments. Our results indicate that lipid droplets and carotenoid droplets are homologous structures and that Plin6 was likely functionalized early in vertebrate evolution for pigment trafficking.

## Results

### Teleost fish contain a unique PLIN variant, Plin6, which arose from the first two vertebrate genome duplications

Our database analysis identified four zebrafish *plin* paralogs. Phylogenetic and conserved synteny analysis indicated that the three human genes *PLIN1, PLIN2,* and *PLIN3* each have clear zebrafish orthologs: *plin1* (XP_003200505.1), *plin2* (NP_001025433.1) and *plin3* (AAH56585.1), respectively. The fourth zebrafish paralog (NP_001103951.1), currently annotated as a 'hypothetical protein (zgc:162150, ENSDARG00000076844.4),' was restricted to a phyologenetic clade that is distinct from the other *PLIN* genes (*Figure 1*). For reasons elaborated below, we call this novel gene *plin6*. Orthologs of *plin6* were identified only in ray fin fish (actinopterygians), including zebrafish (*Danio rerio*), pufferfish (*Tetraodon nigroviridis*; CAF95167.1 and *Takifugu rubripes*; XP_003965836.1), tilapia (*Oreochromis niloticus*; XP_003451283.1) and medaka (*Oryzias latipes*; XP_004078586.1). Importantly, a *plin6* ortholog is present in the spotted gar (*Braasch et al., 2016*), which diverged from the teleost lineage prior to the teleost genome duplication (TGD) about 350 Mya (*Amores et al., 2011*), suggesting an ancient origin of this clade. Finally, as we detail below, systematic phylogenetic and conserved synteny analyses revealed that all existing vertebrate *PLIN* genes can be traced to one of the four products resulting from two genome duplication events that occurred in early vertebrate evolution (VGD1 and VGD2) (*Holland, 1999*; *Dehal and Boore, 2005*; *Nakatani et al., 2007*; *Cañestro et al., 2009*).

### PLIN 1

To obtain a phylogenetic hypothesis for the historical relationships of vertebrate plin genes, we used EnsemblCampara GeneTrees (*Vilella et al., 2009*; *Flicek et al., 2013*) The resulting analysis showed that among lobefin fish (sarcopterygians), *PLIN1* orthologs can be identified in humans and other tetrapods and in the basally diverging coelacanth (*Figure 1*). Among teleosts, a single *PLIN1* ortholog was identified in zebrafish (ENSDARG00000054048), as well as in the spotted gar, a basally diverging rayfin fish. We conclude that the last common ancestor of all bony fish had a *plin1* gene.

### PLIN2

Phylogenetic analysis identified *PLIN2* orthologs in tetrapods, in coelacanth, in numerous teleost species, as well as in spotted gar (*Figure 1*). Although the EnsemblCompara GeneTree analysis suggested that stickleback (*Gasterosteus aculeatus*), Amazon molly, and other percomorphs have an ortholog of the human *PLIN4* gene (ENSGACG00000017870 in stickleback), detailed conserved synteny analysis (*Catchen et al., 2009*, *2011b*) indicates that this percomorph clade gene is embedded in a chromosome segment that is more closely related to human *PLIN2* than it is to human *PLIN4* (*Figure 1—figure supplement 1*). Furthermore, the stickleback ENSGACG00000017870 gene neighborhood has numerous gene duplicates shared with the stickleback chromosomal neighborhood

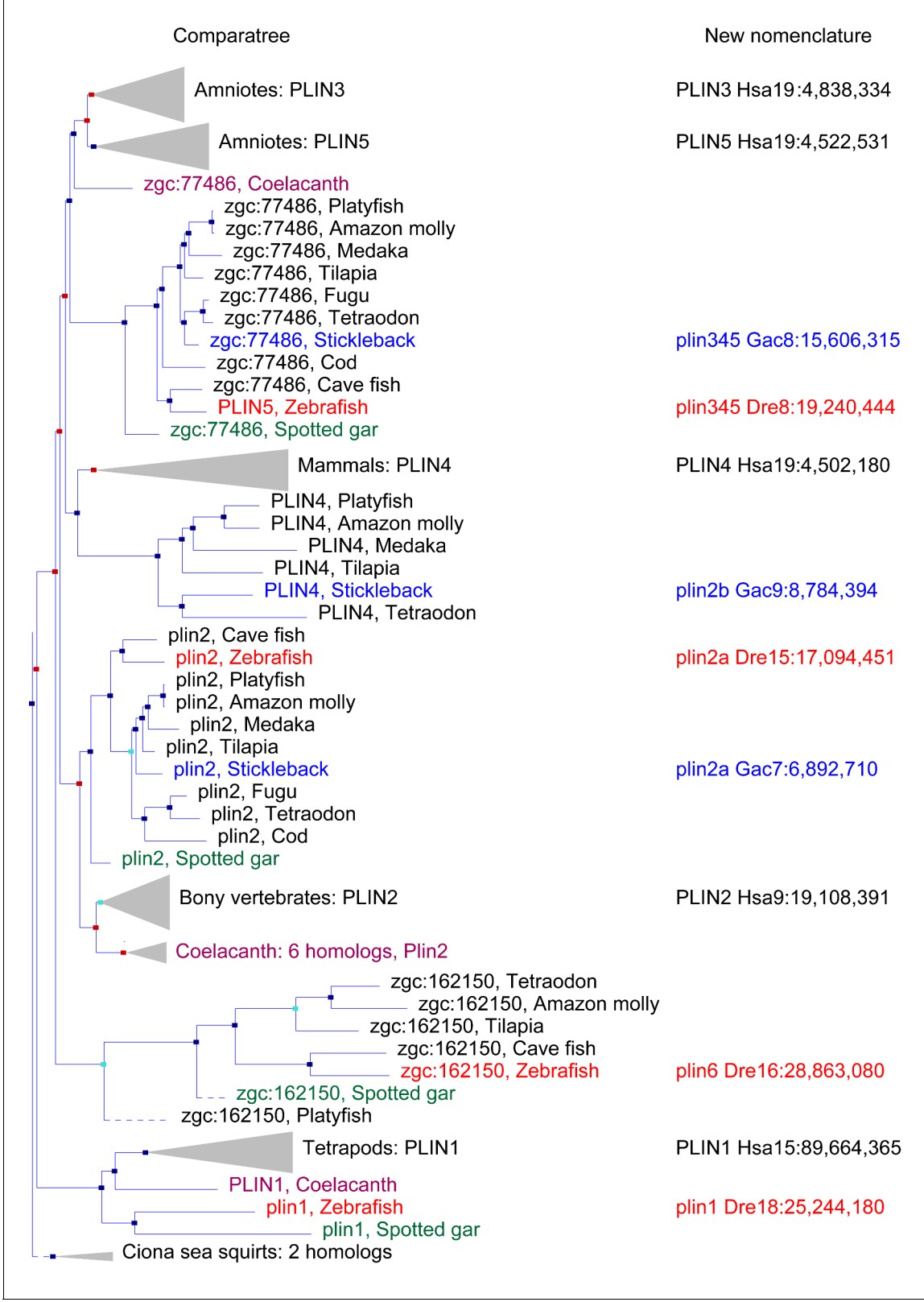

**Figure 1.** A phylogenetic analysis of *Plin*-family genes using EsemblCompara Gene Trees (*Vilella et al., 2009*; *Flicek et al., 2013*). Large clades are collapsed, rayfin fish are shown in detail. Original nomenclature is that of Ensembl with new names assigned for genes of zebrafish and stickleback, along with chromosome locations, to better reflect the historical origins when taking conserved syntenies into account. Source: ENSGT00500000044795.

*Figure 1 continued on next page*

*Figure 1 continued*

The following figure supplements are available for figure 1:

**Figure supplement 1.** Conserved syntenies for the gene Ensembl calls *plin4* in percomorph fish (stickleback gene ENSGACG00000017870), which should be renamed as *plin2b*.

**Figure supplement 2.** A model for the origin of *plin* genes that fits available data.

surrounding the gene Ensembl calls *plin2* (*Figure 1—figure supplement 1*). This analysis shows that stickleback, Amazon molly, and some other percomorphs retained two copies of *plin2* that arose in the teleost genome duplication (*Amores et al., 1998*; *Postlethwait et al., 1999*; *Taylor et al., 2003*; *Jaillon et al., 2004b*); we call these genes *plin2a* (e.g., ENSGACG00000019638) and *plin2b* (e.g., ENSGACG00000017870).

## PLIN3, PLIN4, and PLIN5

In the human genome, *PLIN4*, *PLIN5*, and *PLIN3*, reside in a 15-gene interval, suggesting an origin by tandem duplication (*Figure 2A*). Furthermore, coelacanth, a basally diverging lobefin vertebrate, has a single gene (ENSLACG00000013515) equally related phylogenetically to human *PLIN3* and *PLIN5* (*Figure 1*). Rayfin fish, including teleosts and spotted gar, a basally diverging rayfin, have a single gene that is phylogenetically equally related to the amniote *PLIN3* and *PLIN5* (*Figure 1*). The human chromosome segment containing *PLIN4*, *PLIN5,* and *PLIN3* on *Homo sapiens* chromosome-19 (Hsa19, *Figure 2A–1*) is orthologous to the region of *Danio rerio* chromosome-8 (Dre8, *Figure 2A–2*) that contains the zebrafish *plin5* gene (*Figure 2A*). The presence of a single *plin3/4/5* gene in teleosts (called *plin5* in zebrafish) indicates that *PLIN3/4/5* also originated before the divergence of rayfin and lobefin vertebrates. And the adjacent or near adjacent location of *PLIN4*, *PLIN5*, and *PLIN3* in the human genome suggests and origin by tandem duplication events in the tetrapod lineage after the divergence of coelacanth and human lineages.

Because birds and a lizard have a copy of *PLIN3* and *PLIN5,* but *PLIN4* is present only in mammals (*Figure 1*), we conclude that a *PLIN3/4/5* gene was present in the last common ancestor of tetrapods and was duplicated in stem tetrapods to provide *PLIN3* and *PLIN4/5*, followed by the duplication of *PLIN4/5* to form the adjacent genes *PLIN4* and *PLIN5* in the human lineage after the divergence of mammals from reptiles (including birds).

## PLIN6

Phylogenetic analysis of *PLIN*-family genes identified a rayfin *plin* clade that includes spotted gar and two major teleost clades: ostariophysans, including zebrafish and *Astyanax* cavefish; and percomorphs, including the pufferfish *Tetraodon nigroviridis*, Amazon molly, and tilapia, but excludes lobe fin species (*Figure 1*). Here we rename this *PLIN*-related gene as *plin6*. Tetrapods do not appear to contain orthologs of *plin6* (*Figure 1*).

The phylogenetic distribution of *plin6* genes could be explained either by the hypothesis that *plin6* originated before ray fin/lobe fin divergence, likely in the VGD2 event, but was lost in the human lineage, or alternatively, that *plin6* originated in the ray fin lineage after it diverged from the lobe fin lineage, by tandem duplication or by reverse transcription. These two hypotheses make different predictions for conserved syntenies. The 'VGD2-origin' hypothesis predicts that the chromosome segment containing *plin6* in teleosts would be paralogous to other *Plin* gene-containing chromosome segments. In contrast, the 'ray fin origin' hypothesis predicts that *plin6* should be near another *Plin* gene in the case of an origin by tandem duplication, or in a chromosome segment with no relationship to other *Plin* genes in the case of an origin by reverse transcription. In zebrafish, *plin6* lies on chromosome Dre16 at nucleotide position 28,863,080 (*Figure 2B–1*). This portion of Dre16 shares conserved syntenies with a portion of gar linkage group Loc24 that includes the gar ortholog of *plin6* (*Figure 2B–2*). In turn, this portion of the gar genome is orthologous to a region on human chromosome Hsa1 (*Figures 2B–3*). Comparison of *Figure 2B–2* and B-3 illustrates that chromosome rearrangements in the neighborhood of the gar *plin6* gene: the human orthologs of genes flanking *plin6* (*zgc:171704/FLAD1* to the right and *syt11b/SYT11* to the left are more than 50

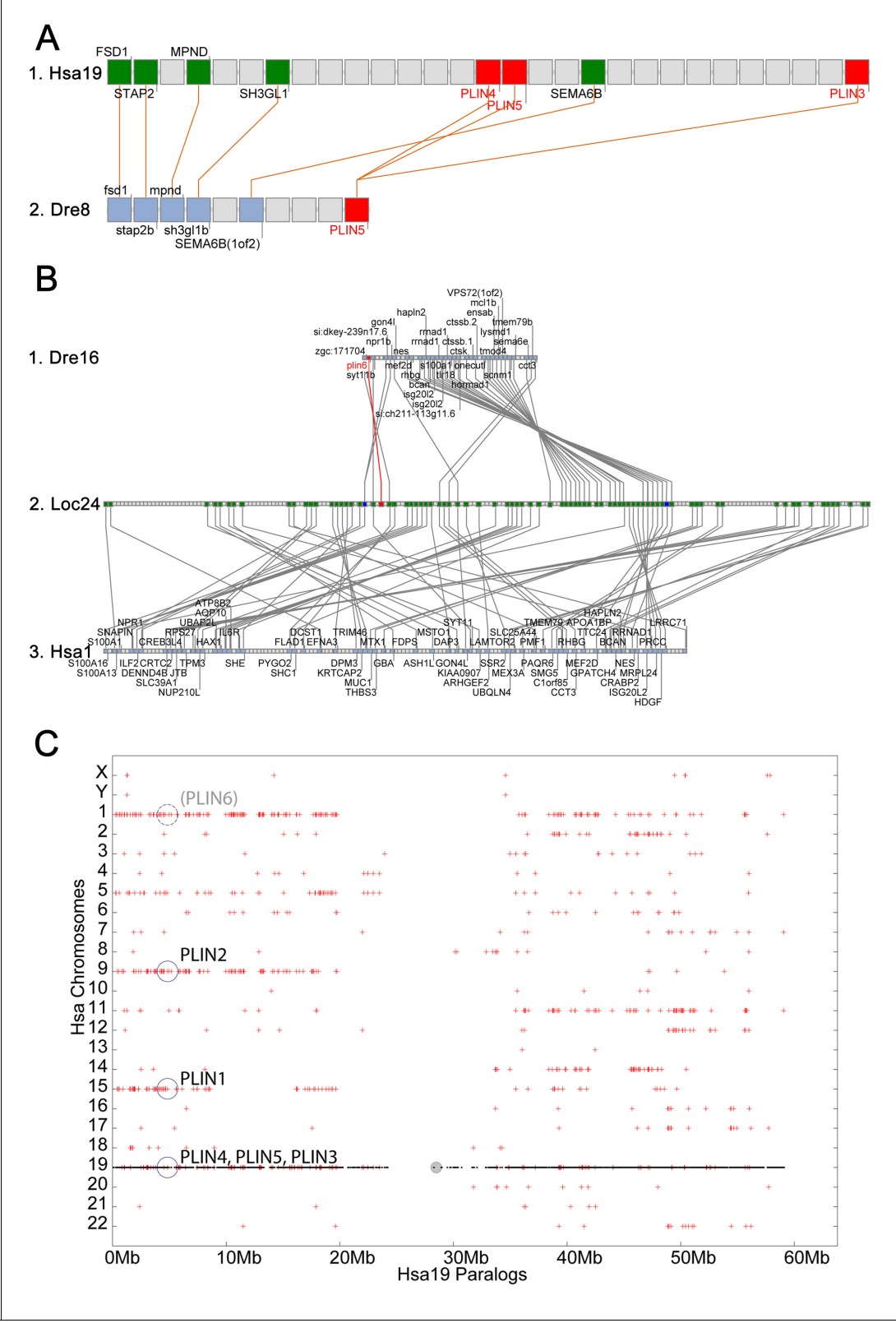

**Figure 2.** Conserved syntenies for *PLIN3/4/5* in human and zebrafish and *plin6* in zebrafish, gar, and human. (**A**) Conserved synteny analysis shows that the chromosome segment containing zebrafish 'PLIN5' (Dre8) is orthologous to the human chromosome segment containing *PLIN4, PLIN5,* and *PLIN3* (Hsa19). (**B**) The zebrafish *plin6* lies on chromosome Dre16 (*Amores et al., 2011*), which shares conserved syntenies with a portion of gar linkage group Loc24 that includes the gar *plin6* (*Amores and Force, 1998*). This portion of the gar genome is orthologous to a region on human chromosome Hsa1

*Figure 2 continued on next page*

*Figure 2 continued*

(C), but chromosome inversions distort the mapping of the human chromosome to the gar genome (follow grey lines from **B** to **C**) and no *PLIN*-related gene appears. (C) Dotplot analysis showing paralogs of Hsa19 genes on the other 22 human chromosomes directly above their position on Hsa19 (*Catchen et al., 2009*, *2011b*). The region of Hsa1 that is predicted to be orthologous to the *plin6*-containing region of zebrafish and gar is circled. It is in the portion of Hsa1 predicted to be paralogous to the regions of Hsa9, Hsa15, and Hsa19 that contain the *PLIN2, PLIN1,* and *PLIN4-PLIN5-PLIN3* loci, respectively.

The following figure supplements are available for figure 2:

**Figure supplement 1.** Protein sequence comparison of zebrafish Plin paralogs.

**Figure supplement 2.** Protein sequence comparison of Plin1 and Plin6 in selected mammals and teleosts.

**Figure supplement 3.** Representation of the conservation of the three conserved PKA sites found in the C-terminus of zebrafish Plin6 (*D. rerio*).

genes apart on Hsa1 and no *PLIN*-related gene appears near the expected location. This result indicates that chromosome rearrangements in the lobefin lineage likely led to the destruction of the *plin6* ortholog. We conclude that the chromosome region containing *plin6* was retained in the ray fin and human lineages but that *plin6* appears only in the ray fin lineage, a result compatible with the VGD2-origin hypothesis or the retrotransposition hypothesis but not the tandem duplication mechanism for origin of *plin6.*

If *plin6* comes from VGD2, then it should occupy—with other *Plin* genes—ohnologous chromosome segments in both teleosts and tetrapods. (Note that paralogs that derive from genome duplication events are special and thus are given the name 'ohnolog', in honor of Susumu Ohno, who first

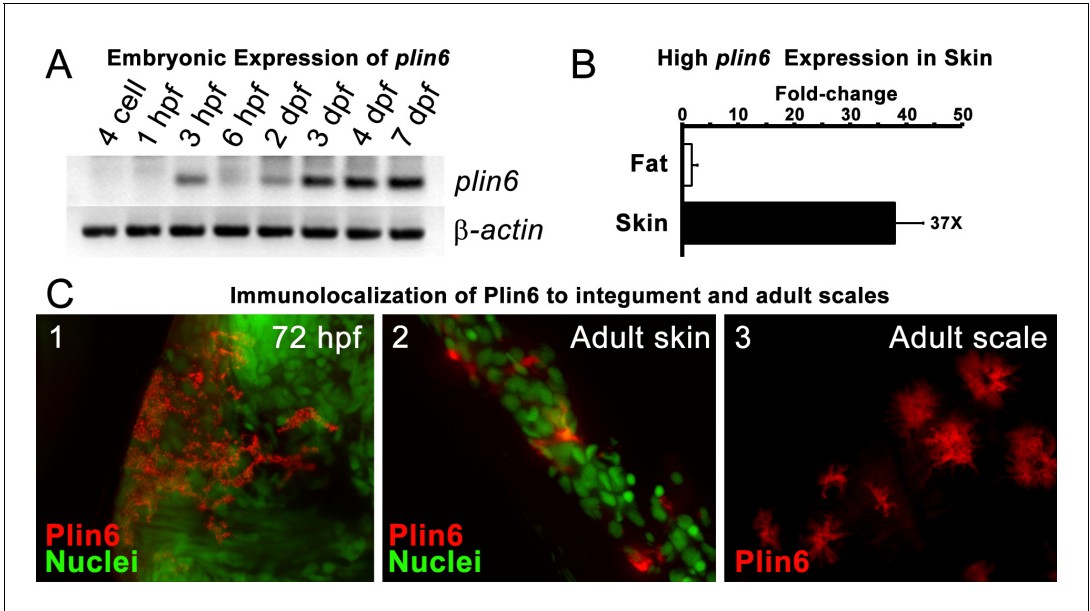

**Figure 3.** Plin6 is expressed in early zebrafish development and in adult skin. (A) Transcripts of *plin6* were detected at various stages of early zebrafish development. Hours post-fertilization (hpf); days post-fertilization (dpf). (B) Graph depicting expression levels of *plin6* in the isolated fat and skin as detected by real-time PCR. Expression of *plin6* in skin is 28x more than in fat. (C) Immunolocalization of Plin6 to the integument in wholemount embryos at 72 hpf (C1), in sectioned adult skin (C2), and in isolated whole scales (C3). The stellate expression of Plin6 in adult scales is consistent with the morphology of xanthophores located within the scale.

The following figure supplement is available for figure 3:

**Figure supplement 1.** Plin6 expression in tissue samples from skin, muscle, liver, and gut.

promoted the idea that vertebrate genomes arose by genome duplication (*Ohno, 1970*; *Wolfe, 2000*). Evidence for the origin of the four original vertebrate *PLIN* genes (*Plin1, Plin2, Plin3/4/5*, and *plin6*) as VGD2 ohnologs comes from examining paralogons in the human genome. The region of Hsa1 that is predicted to be orthologous to the *plin6*-containing region of zebrafish and gar (*Figures 2B–3*) is in the portion of Hsa1 predicted to be paralogous to the regions of Hsa9, Hsa15, and Hsa19 that contain the *PLIN2*, *PLIN1*, and *PLIN4-PLIN5-PLIN3* loci, respectively (*Figure 2C*). These portions of human chromosomes Hsa1, Hsa9, Hsa15, and Hsa19 were previously shown to be derived from the VGD1 and VGD2 events (*Dehal and Boore, 2005*; *Nakatani et al., 2007*; *Cañestro et al., 2009*) as predicted by the hypothesis that *plin6* originated in the second round of vertebrate genome duplication. Therefore, we conclude that *plin6* is an ohnolog derived from the VGD1 and VGD2 events that has gone missing in the lobefin lineage. A detailed history of the *PLIN* gene family, including individual gene duplications and whole genome duplications, is presented (*Figure 1—figure supplement 2*).

## Conserved properties of Plin6

Zebrafish *plin6* mRNA encodes a 490 amino acid residue protein that contains an N-terminal PAT domain (*Figure 2—figure supplement 1*) and three consensus sites for protein kinase A (PKA) phosphorylation in the C-terminal region (*Figure 2—figure supplement 1*). The overall sequence similarity of Plin6 with the other zebrafish Plin paralogs ranged from 29–34%, and was largely limited to the conserved N-terminal PAT domain (*Figure 2—figure supplements 1–2*). Importantly, the three PKA sites that were found in the C-terminus of zebrafish Plin6 were conserved in pufferfish, tilapia, medaka, goldfish (*Figure 2—figure supplement 3*), with the final PKA site conserved in spotted gar (*Figure 2—figure supplement 3*), strongly indicating that they are ancient and functionally significant.

## Plin6 is expressed in xanthophores and traffics with carotenoid droplets

During early development, expression of *plin6* was strong in 3 days post-fertilization (dpf) embryos and continued at least into larval stages (*Figure 3A*). In the adult, contrary to the expectation that *plin6* would be expressed in fat cells, real-time PCR showed abundant expression of *plin6* in skin, where we detected *plin6* expression levels to be 37-fold higher than that observed in adipose tissue (*Figure 3B*). To determine the specific cellular localization of Plin6 in fish integumentum, we generated affinity-purified antibodies to recombinant full-length Plin6 and performed immunohistochemical analysis. We confirmed that Plin6 immunolocalized to the integumentum in 72 hr post-fertilization (hpf) embryos (*Figure 3C–1*), in histological sections of adult skin (*Figure 3C–2*) and scales (*Figure 3C–3*), and that Plin6 was not expressed in adult muscle, liver, or gut tissue (*Figure 3—figure supplement 1*). The stellate morphology of the position of the protein in adult scales is consistent with Plin6 immunolocalization to xanthophores, which we later more definitively confirmed.

Xanthophores are chromatophores that concentrate yellow/orange pigment in poikilothermic vertebrates such as amphibians, reptiles, and fish, but not birds or mammals (*Kimura et al., 2014*; *Parichy and Spiewak, 2015*). In teleost fish, xanthophores contain yellow pteridine pigments stored in pterinosomes (*Matsumoto, 1965*; *Matsumoto and Obika, 1968*; *Khoo and Phang, 1992*, *2012*), and red/orange carotenoid pigments stored in carotenoid droplets (*Goodrich et al., 1941*; *Matsumoto, 1965*). Pterinosomes and CD can be readily distinguished by numerous criteria; for example, CD possesses hydrophobic carotenoids that disperse from a perinuclear aggregate in response to PKA activation and fluoresce when excited by blue light (*Goodrich et al., 1941*; *Matsumoto, 1965*; *Bagnara and Hadley, 1969*; *Lynch et al., 1986a*, *1986b*; *Khoo and Phang, 1992*; *Kimler et al., 1993*). We observed a dense aggregation of CD in the perinuclear region of zebrafish xanthophores under basal conditions (*Figure 4B*), as found in goldfish xanthophores (*Figure 4—figure supplement 1*), which have served as a model system for analysis of pigment trafficking (*Lynch et al., 1986a*, *1986b*; *Kimler et al., 1993*). As expected for CDs, these aggregates were fluorescent when excited by blue light (*Figure 4A', B'*), and rapidly dispersed following activation of PKA with forskolin/IBMX (*Figure 4C,C'*).

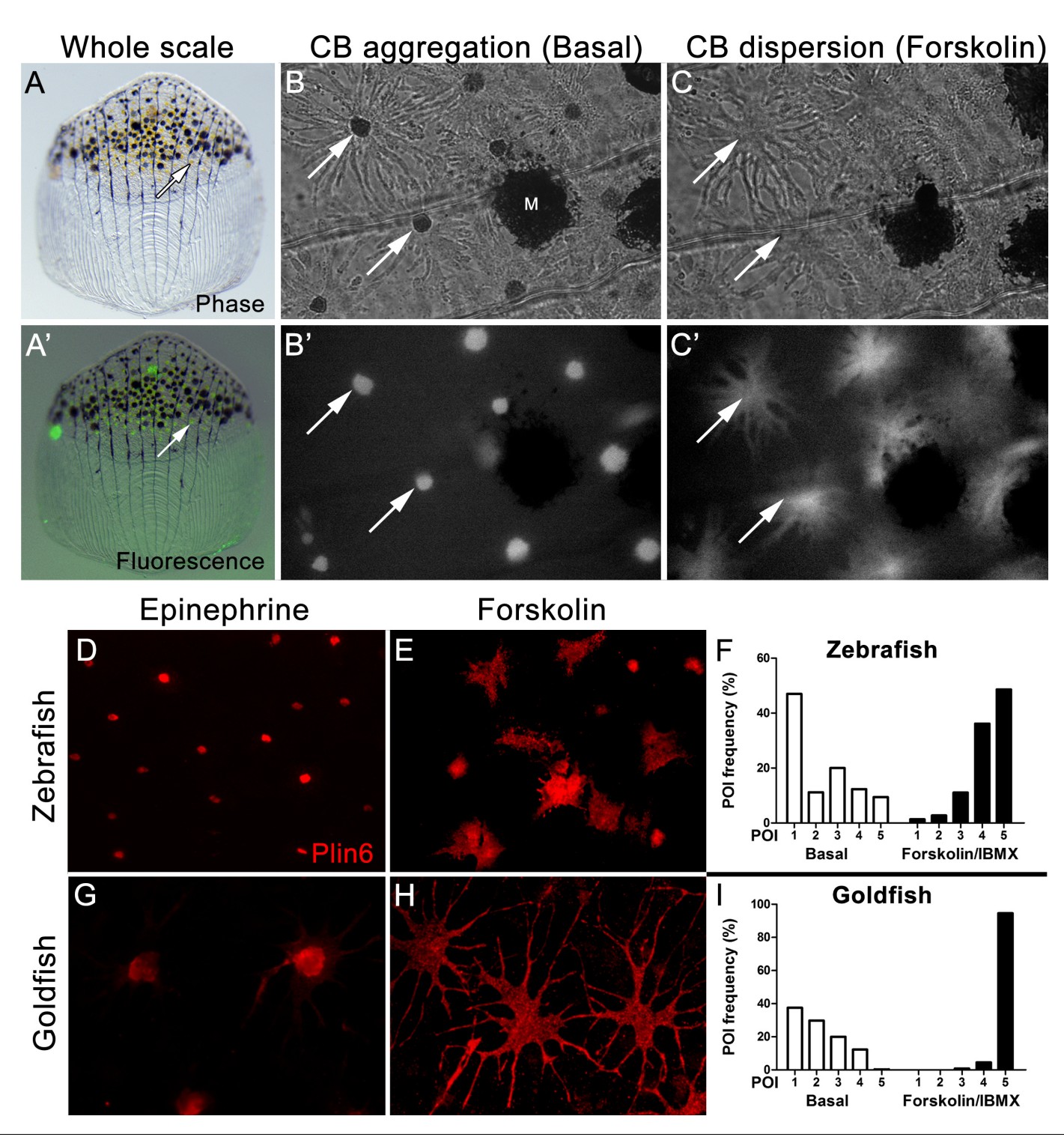

**Figure 4.** Carotenoid pigment aggregation and dispersion in adult zebrafish xanthophores correlates to Plin6 expression. (A) An isolated scale containing melanophores (black) and xanthophores (gold, arrow). (A') Fluorescent and brightfield overlay showing xanthophore auto-fluorescence (the two large fluorescent spots are auto-fluorescent debris). (B) Higher magnification image showing many star-shaped xanthophores (arrows) in a basal state of pigment aggregation. A few melanophores (M) can also be visualized. (B') The fluorescence image of B showing the auto-fluorescence in xanthophores. (C) Following a 1 hr incubation in forskolin, pigment is dispersed. (C') The fluorescence image shows the pigment dispersion. ( D–E) Plin6 is immunolocalized to xanthophores in isolated scales from zebrafish in both aggregated (D; Epinephrine) and dispersed states (E; Forskolin). (F) Quantification of the percentage of xanthophores at each Pigmentary Organelle Index (POI) normalized to 100% (p<0.0001). (G–H) Plin6 is

*Figure 4 continued on next page*

*Figure 4 continued*
immunolocalized to xanthophores in isolated scales from goldfish in both aggregated (**G**) and dispersed states (**H**). (**I**) Quantification of the percentage of xanthophores at each Pigmentary Organelle Index (POI) normalized to 100% (p<0.0001).
The following source data and figure supplement are available for figure 4:

**Source data 1.** Relative frequency distribution of Pigment Organelle Index under basal conditions and after treatment with forskolin/IBMX in Zebrafish (*Figure 4F*).
**Source data 2.** Relative frequency distribution of Pigment Organelle Index under basal conditions and after treatment with forskolin/IBMX in Goldfish (*Figure 4G*).
**Figure supplement 1.** Thin section transmission electron microscopy (TSTEM) of zebrafish and goldfish xanthophores.

Parallel analysis of Plin6 immunolocalization in both zebrafish and goldfish xanthophores demonstrated that Plin6 was targeted to the perinuclear region under basal conditions or following treatment with epinephrine, which inhibits PKA activation (*Figure 4D,G*). In contrast, activation of PKA with forskolin-IBMX led to rapid and extensive dispersion of Plin6 immunofluoresence throughout the xanthophore soma in both species (*Figure 4E,H*). Evaluation of aggregation and dispersion in xanthophores using the semiquantitative Pigmentary Organelle Index (*Taylor et al., 1991*) demonstrated that forskolin-IBMX treatment dramatically and significantly increased the dispersion of CD and Plin6 immunofluorescence in both species (p<0.0001; *Figure 4F,I*).

## Plin6 fractionates with purified CD and is phosphorylated under conditions that disperse pigment

We purified CD by flotation of skin homogenates through a sucrose cushion and found that Plin6 immunoreactivity was highly enriched the orange pigmented floating fraction (*Figure 5A*). Immunoblotting of the CD fraction with anti-Plin6 identified a major band of 54.5 kDa, predicted by the conceptual translation of *plin6* mRNA. We also observed variable levels of a 39.4 kDa band that appears to be a proteolytic fragment because its abundance increased with prolonged incubation (not shown).

Plin6 contains three predicted PKA phosphorylation consensus sites that are phylogenetically conserved among Plin proteins (*Figure 2—figure supplements 2–3*), suggesting that phosphorylation might correspond to PKA-mediated CD trafficking. To test whether PKA activates Plin6 by phosphorylation, we isolated protein from zebrafish scales following treatment with epinephrine or forskolin-IBMX to inhibit or stimulate PKA activity, respectively. Proteins were resolved using polyacrylamide gels containing Phos-tag, which retards the electrophoretic mobility of phosphorylated proteins (*Kinoshita et al., 2006*). Immunoblot analysis of Phos-tag gels indicated a mixture of the cleavage product, unphosphorylated full-length protein, and phosphorylated Plin6 following epinephrine treatment (*Figure 5B*). Activation of PKA by forskolin-IBMX, which promotes CD dispersion, strongly shifted most of the full length Plin6 to a low mobility band, indicating a maximally-phosphorylated state (*Figure 5B*).

## Plin6 targets to the surface of CD

The above data indicate that the trafficking of Plin6 corresponds closely with that of CD, suggesting that Plin6 might be specifically targeted to these structures. Previous analysis of goldfish xanthophores indicated that CD are ~50 nm in diameter (*Kimler et al., 1993*), which is too small to resolve by optical microscopy. Thin-section TEM of zebrafish scales indicated that zebrafish CDs are ~25 nm in diameter in both the aggregated and dispersed states (*Figure 4—figure supplement 1*). Importantly, immunoTEM analysis of zebrafish skin xanthphores demonstrated that Plin6 is heavily targeted to the surface of individual CD and form characteristic knob-like substructures on the larger carotenoid body (*Figure 5C*).

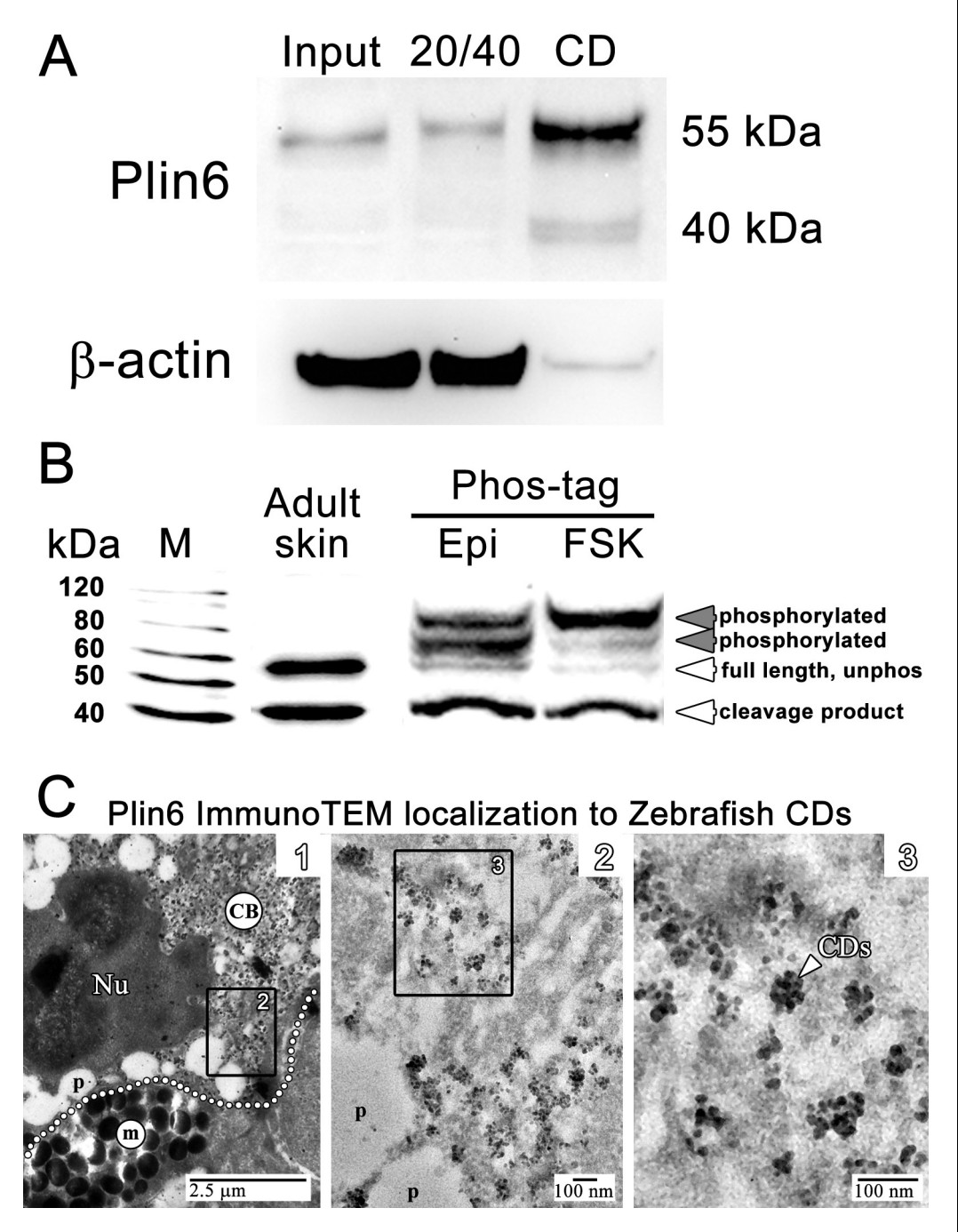

**Figure 5.** Plin6 is localized to carotenoid droplets and is phosphorylated under conditions that disperse pigment. (**A**) Western blot analysis of Plin6 expression from three fractions of skin homogenate separated by a sucrose gradient; total protein homogenate in 40% sucrose prior to centrifugation (Input), the protein fraction at the interface of the 20% and 40% sucrose layers following centrifugation (20/40), and the purified CD fraction that floated above the 20% sucrose following centrifugation (CD). (**B**) Western blot analysis of Plin6 expression from untreated skin (adult skin), and from scales treated with epinephrine (Epi) or forskolin-IBMX (FSK) to inhibit or stimulate PKA activity, respectively. Proteins from treated scales were resolved using polyacrylamide gels containing Phos-tag, which retards the electrophoretic mobility of phosphorylated proteins. Plin6 is largely unphosphorylated (Unphos, white arrowheads) under conditions that promote aggregation (Epi) and highly phosphorylated (Phos, grey arrowheads) under conditions that promote dispersion (FSK). (**C**) Plin6 immunoTEM in zebrafish scales. (**C-1**) A dotted white line demarks the rough boundary between a xanthophore (top) and

*Figure 5 continued on next page*

*Figure 5 continued*

melanophore (bottom). The carotenoid body (CB) with clusters of CD puncta is located in a field adjacent to the nucleus (Nu) of the xanthophore. Pterinosomes (p) within the xanthophore are also present, as well as a field of melanosomes (m) within the adjacent melanophore. (C-2) Higher magnification of the boxed inset shown in C-1. (C-3) Higher magnification of the boxed inset shown in C-2 show Plin6 immunoreactivity targeted to clusters of CDs (arrowhead).

## Plin6 is not required for xanthophore formation but promotes the concentration of pigment within xanthophore carotenoid droplets and the aggregation of carotenoid bodies

To identify the function of Plin6, we generated a stable line of *plin6* knockout fish using TALENS. Western blot and immunohistochemical analysis confirmed the absence of Plin6 protein in the mutant animals (*Figure 6—figure supplement 1*). Knockout of *plin6* resulted in no apparent systemic defects in mutant embryos, larvae, or adults and had no obvious effect on the development and location of xanthophores in the body stripes or scales (*Figure 6—figure supplement 2*). However, *plin6* mutants exhibited a striking reduction in the amount of carotenoid pigment within the xanthophores (*Figure 6A–D*). Analysis of carotenoid levels from whole skin extracts confirmed that knockout of *plin6* reduced skin carotenoid levels by more than 50% (*Figure 6E,F*, p<0.001). In addition, knockout of *plin6* reduced integumentary TAG levels by nearly 40% (*Figure 6G*, p<0.02).

Examination of the residual carotenoid fluorescence in isolated scales (by increasing camera exposure time) indicated that *plin6* mutants were unable to tightly aggregate CD within xanthophore cells in response to inhibition of PKA with epinephrine (*Figure 7B,C*). Surprisingly, knockout of *plin6* did not affect dispersion of CD into cytoplasmic processes following PKA activation by forskolin/IBMX (*Figure 7B'*). These data show that Plin6 is required for CD aggregation, but not full CD dispersion from an already relaxed state.

## Plin6 targets mammalian LD, suggesting that CD and LD are homologous structures

CD are thought to be unique subdomains of the smooth endoplasmic reticulum (SER) that forms between membrane leaflets and thus contains hydrophobic cargo (carotenoid pigment and TAG) surrounded by a phospholipid monolayer (*Kimler et al., 1993*). Similarly, mammalian lipid droplets (LD) are thought to originate between membrane leaflets of the SER (*Brasaemle and Wolins, 2012*) and after budding from the SER contain hydropobic cargo (mainly triglyceride and cholesterol esters surrounded by a phospholipid monolayer). To test whether Plin6 can target mammalian LD, we expressed fluorescently-tagged zebrafish Plin6 in lipid-loaded COS7 cells in the absence and presence of mouse PLIN1 (*Figure 8*). We found that zebrafish Plin6 (*Figure 8A,C*) accumulated on the surface of LD labeled with LipidTOX when expressed alone (*Figure 8B,C*). Moreover, when Plin6 was expressed in mammalian COS7 cells, it was strongly targeted to LD containing mouse PLIN1 (*Figure 8D–F*), indicating that CD and LD are homologous structures.

## Discussion

PLINs comprise an evolutionarily conserved superfamily that plays a key role in cell-specific functions of LD (*Miura et al., 2002*). Phylogenetic analysis indicates that the vertebrate *PLIN* gene family arose by the duplication of a single gene present in a chordate ancestor (represented today by a gene in the urochordate *Ciona* (ENSCING00000004283)) in two rounds of whole genome duplication at the base of the vertebrate radiation giving four genes: *PLIN1, PLIN2,* the ancestor of the *PLIN4-PLIN5-PLIN3* cluster, and *plin6.* After the divergence of rayfin and lobefin vertebrates, *plin6* went missing in the lobefin lineage but was retained in the rayfin lineage. In the lobefin lineage, one of the *PLIN* genes experienced a tandem duplication in the tetrapod lineage after it diverged from the coelacanth lineage producing *PLIN3* and *PLIN5*. Subsequently, after the divergence of sauropsids (birds and 'reptiles') from synapsids (mammals and mammal-like 'reptiles'), *PLIN5* underwent a tandem duplication, leaving *PLIN5* and *PLIN4*, completing the five-gene repertoire of the human *PLIN* family. Meanwhile, in the rayfin lineage, after the teleost genome duplication, the four *plin*-family genes

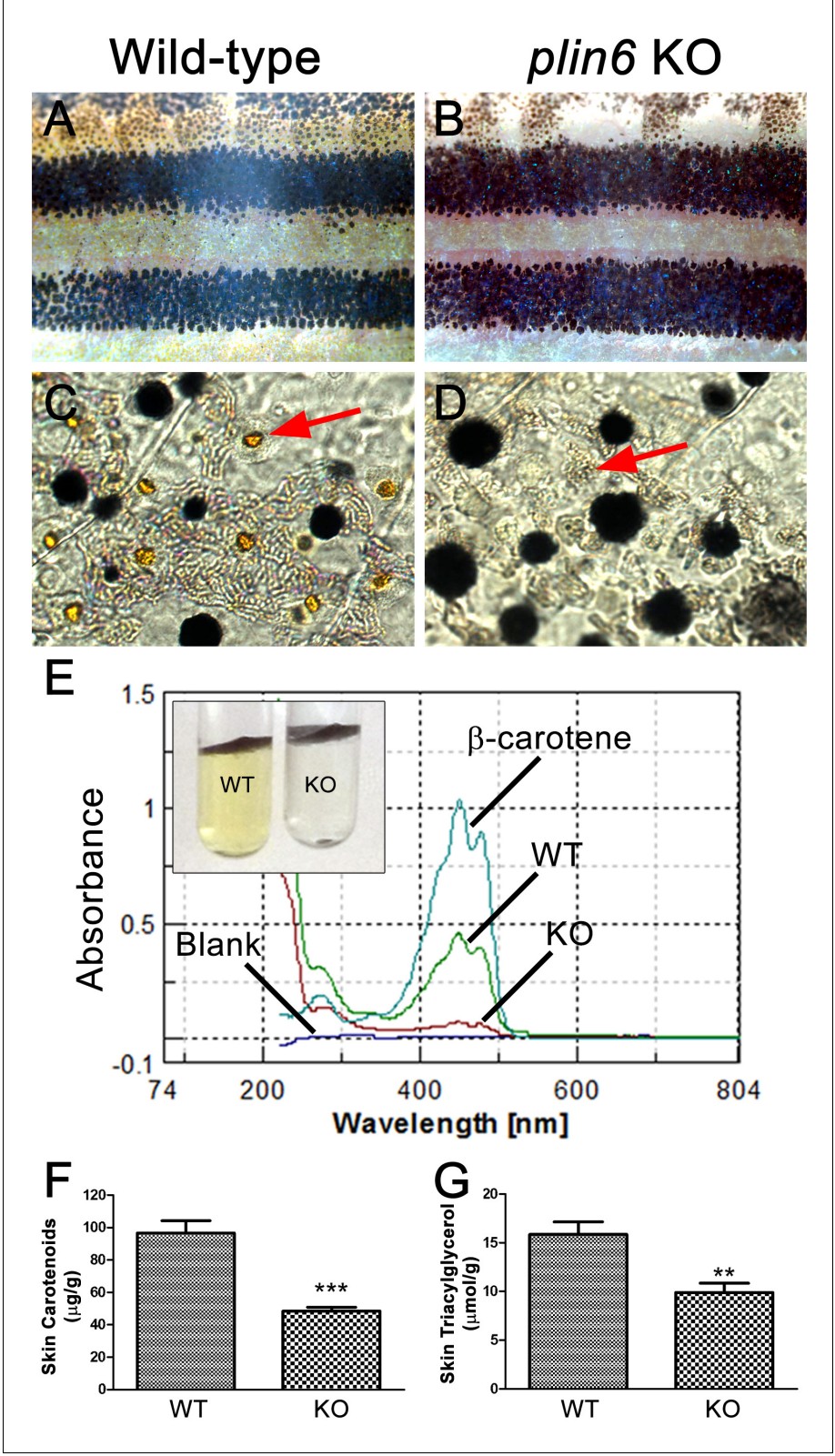

**Figure 6.** Reduced carotenoid concentration in *plin6* knockout zebrafish. Brightfield images of the flank and isolated scales from adult wild-type (**A, C**) and *plin6* mutants (**B, D**) show a reduction in yellow carotenoid pigmentation in mutant tissue (red arrows in **C, D**). (**E**) Extraction of carotenoid pigment from wild-type and *plin6* mutant integument (inset) and the corresponding absorbances of carotenoids from a *β*-carotene standard, wild-

*Figure 6 continued*

type skin (WT), and *plin6* mutant skin (KO). (F) Quantification of skin carotenoid levels from wild-type (WT) and *plin6* mutant skin (KO). N = 4; *** = p<0.001. (G) Quantification of skin triacylglycerol levels from wild-type (WT) and *plin6* mutant skin carotenoid droplets (KO). N = 3; ** = p<0.02.

The following source data and figure supplements are available for figure 6:

**Source data 1.** Skin carotenoid content of wild type and plin6 knockout fish (*Figure 6F*).

**Source data 2.** Skin triacylglycerol content of wild type and plin6 knockout fish (*Figure 6G*).

**Figure supplement 1.** The absence of Plin6 expression in *plin6* mutants.

**Figure supplement 2.** Developing and adult *plin6* mutant fish show no obvious defects.

---

became eight genes that then reverted to single copy except for *plin2*, which some percomorphs maintained as *plin2a* and *plin2b*. This process left the zebrafish with four *plin* genes, one copy of each of the original four that resulted after the vertebrate genome duplication events. These observations point to an ancient function of Plin6 in fish that was lost in tetrapod descendants.

Immunohistochemical analysis demonstrated that Plin6 is highly expressed in fish integumentum, but not in adipose or liver, which are potential sites of neutral lipid storage (*Figure 3* and data not shown). The conservation of the amphipathic N-terminal PAT domain suggested that Plin6 would likely be targeted to LD or LD-like structures, yet its function might be unrelated to TAG storage and mobilization. Furthermore, the C-terminal region of Plin6 contains highly conserved consensus sites for phosphorylation by PKA. The placement of these PKA sites near the C-terminus is reminiscent of the PKA sites in mammalian PLIN1 that are critical for controlling its interactions with lipases and lipase activators (*Subramanian et al., 2004*; *Marcinkiewicz et al., 2006*; *Miyoshi et al., 2007*; *Granneman et al., 2009*); however, the regulatory sequences of PLIN1 likely evolved independently, because no significant amino acid similarity exists between the mammalian and teleost proteins in this region and in addition, PKA sites in Plin1 are not well conserved among lobefin descendants. Together, these results indicate that Plin6 mediates conserved functions that are specific to certain teleosts and likely involve PKA modulation of hydrophobic cargo trafficking.

We found that Plin6 is highly expressed in xanthophores and is specifically targeted to CD, which are involved several adaptive functions, such as cryptic coloration, conspecific recognition, and sexual selection (*Price et al., 2008*). Carotenoid-based pigmentation is found in both vertebrates (*Cote et al., 2010*; *Walsh et al., 2012*) and invertebrates (*Yang et al., 1999*; *Messenger, 2001*), and substantial evidence indicates that CD and LD are homologous structures with a shared evolutionary history. Indeed, fish CD were once referred to as 'lipophores' to emphasize the lipid content of these intracellular organelles (*Bagnara, 1966*). Furthermore, ultrastructural investigations by Taylor and colleagues (*Kimler et al., 1993*) demonstrated that CD, like LD, contain a hydrophobic core surrounded by a phospholipid monolayer. Teleost CD are thought to be a constitutive subdomain of the SER (*Kimler et al., 1993*), whereas mammalian and yeast LD are thought to form in a specialized SER subdomain that is then released as independent LD (*Goodman, 2009*; *Brasaemle and Wolins, 2012*). Mammalian LD that concentrate carotenoids (specifically retinylesters) contain PLIN proteins (*Higashi and Senoo, 2003*; *Straub et al., 2008*; *Orban et al., 2011*), and we show that zebrafish Plin6 and mouse PLIN1 target the same LD structures in mammalian cells. It has been previously shown that mammalian PLIN paralogs and splice variants can differentially concentrate cholesterol ester versus triglyceride within LD structures (*Hsieh et al., 2012*). The mechanisms involved in selective sequestration of hydrophobic cargo are not understood, but could involve cell-specific transport, PLIN-interacting proteins, and the composition of the LD/CD phospholipid monolayer.

The functional evolution of PLIN6 is somewhat a matter of speculation. Mechanisms for concentrating carotenoid pigments are found widely in vertebrates and invertebrates (*Jouni and Wells, 1993*; *Tabunoki et al., 2004*), and thus likely predate the appearance of bony fish. Because fish obtain carotenoids only from their diet, the pigmentation carotenoids confer is thought to represent an 'honest' index of foraging ability (*Lindström et al., 2009*; *Pike et al., 2010*). In addition, the

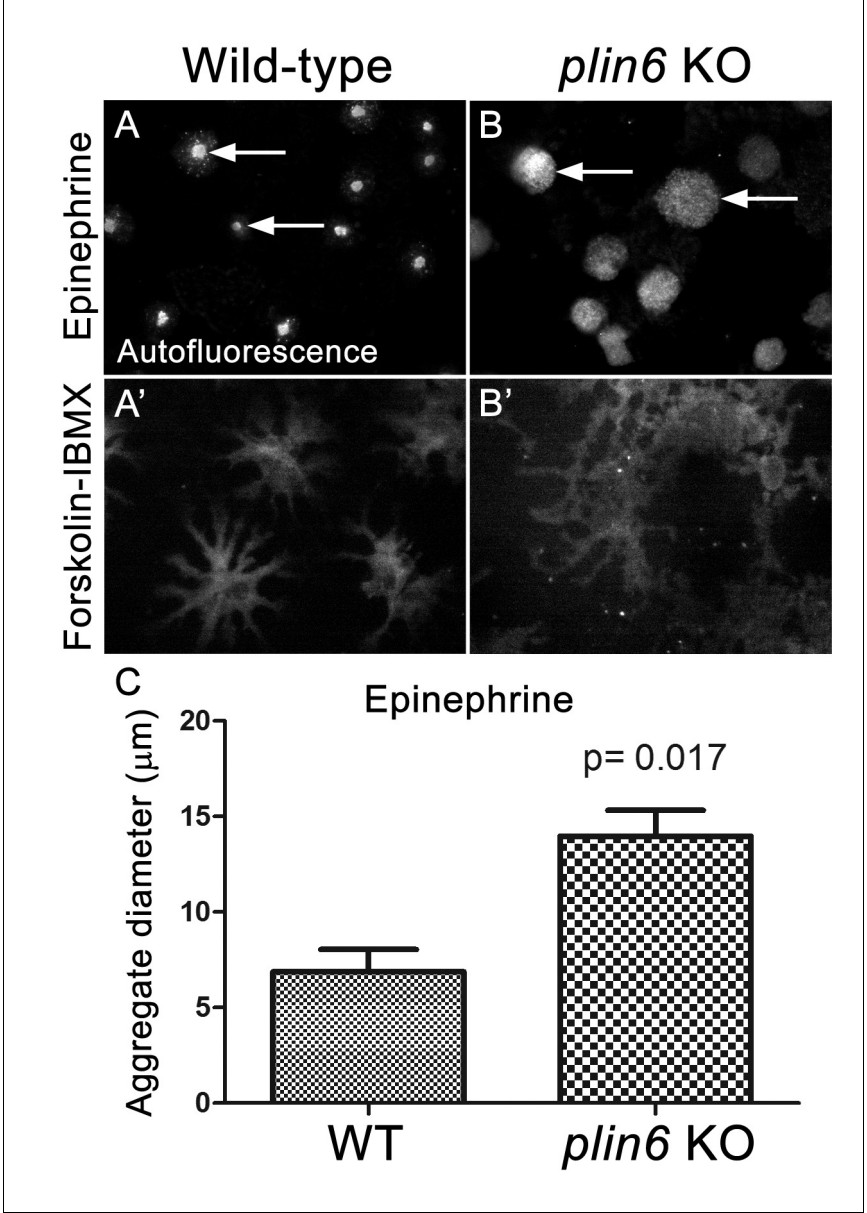

**Figure 7.** Loss of Plin6 function impairs CB aggregation in adult zebrafish xanthophores. (**A–A'**) An isolated scale from a wild-type animal showing xanthophore carotenoid body auto-fluorescence. 1 hr incubation in Epinephrine (**A**) or forskolin-IBMX (**A'**) induced CB aggregation and dispersion, respectively. (**B–B'**) An isolated scale from a *plin6* mutant animal showing xanthophore carotenoid body auto-fluorescence. 1 hr incubation in Epinephrine (**B**) failed to induce tight CB aggregation (arrows). However, incubation in forskolin-IBMX (**B'**) induced CB dispersion. (**E**) Quantification of the aggregate diameter of wild-type and *plin6* mutant xanthophores from measurements of Epinephrine-treated scales (N = 3; p=0.017).

The following source data is available for figure 7:

**Source data 1.** Carotenoid body diameters of wild type and plin6 knockout fish following epinephrine-induced aggregation (*Figure 7C*).

antioxidant effects of carotenoids have been linked to increased male fertility in stickleback fish (*Pike et al., 2010*). Our knockout data demonstrate that expression of Plin6 increases the ability of xanthophores to concentrate pigment, and thus could intensify a preexisting signal for sexual

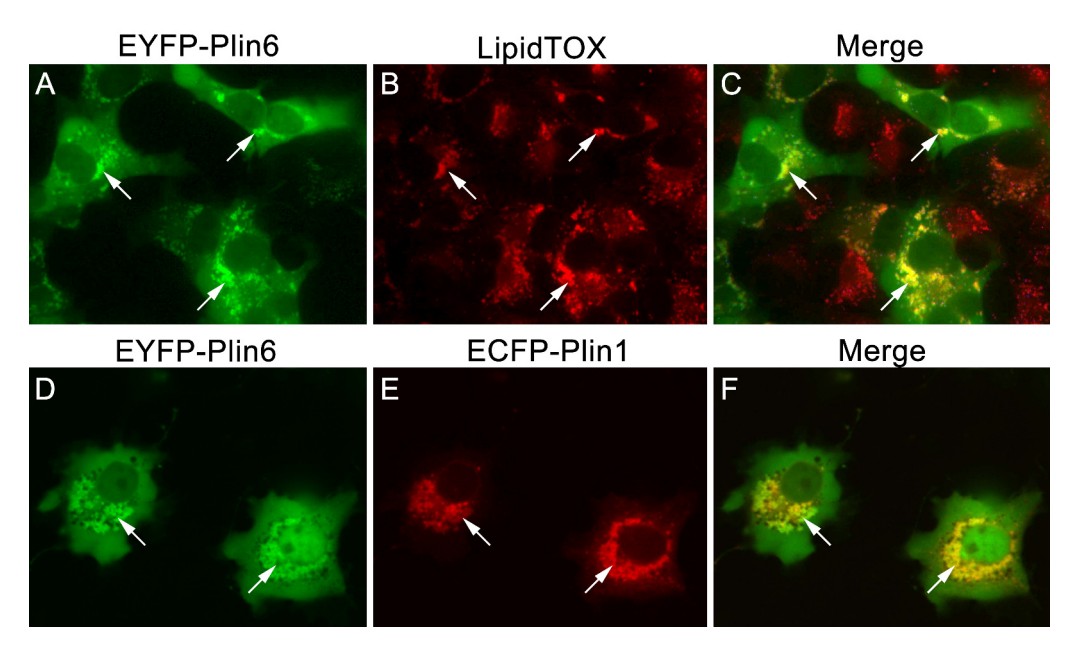

**Figure 8.** CD and LD are analogous structures. (**A**) Expression of EYFP-Plin6 fusion protein in COS7 cells that have been induced to create lipid droplets. Strong expression is observed near the nuclei (arrows). (**B**) Lipidtox staining of intracellular lipid droplets (arrows). (**C**) Overlay of **A** and **B**, showing co-labeling of EYFP-Plin6 fusion and LipidTOX stained lipid droplets (arrows). (**D**) Expression of EYFP-Plin6 fusion protein in COS7 cells (arrows). (**E**) Mouse Plin1-ECFP fusion expression in the same cells shown in panel **A** (arrows). (**F**) Overlay of panels **A** and **B**, showing co-expression of zebrafish EYFP-Plin6 and mouse Plin1-ECFP (arrows).

selection. Dynamic trafficking of melanosomes is found in several cell types and may be important in cryptic coloration (*Bagnara and Hadley, 1969*; *Price et al., 2008*). Because melanosomes and xanthophores derive from a common neural crest progenitor (*Parichy and Spiewak, 2015*), it seems likely that the basic machinery for PKA-induced trafficking predates evolution the of PKA phosphorylation sites in Plin6. With the possible exception of certain amphibians, lobefins (sarcopterygians) lack either xanthophores in their entirety or the ability to aggregate and disperse CDs (*Bagnara and Hadley, 1969*). Therefore, the aggregation and dispersion of CDs across species is correlated with the presence of Plin6. Clearly, the function of Plin6 in CD trafficking would be lost in vertebrates lacking scales and/or carotenoid-based skin pigmentation. In this regard, mammals lack both xanthophores (*Parichy and Spiewak, 2015*) and, as we show here, Plin6.

Finally, our results clearly indicate that CD and LD are homologous structures. A common feature of all PLIN proteins is the ability to target structures with hydrophobic cargo surrounded by a phospholipid monolayer. Diversification and specialization of Plin paralogs throughout evolution seem to have been driven by cell-specific expression and interactions with specific binding partners (*Granneman et al., 2007*, *2009*, *2011*). In this regard, Plin6 appears to have unique properties, including selective expression in xanthophores, the ability to influence the organization of CD, the ability to concentrate specific cargo, and the potential to interact with trafficking machinery. We anticipate that further analysis of Plin6 will inform our understanding of LD, PLIN protein biology, and the evolutionary mechanisms giving rise to the wonderful color variations that are so important for the biology of teleost fish, half of all vertebrate species.

# Materials and methods

## Ethics statement for research and maintenance of zebrafish and goldfish

Wild-type (*AB* strain; Research Resource Identifier: ZIRC_ZL1) and *plin6* mutant zebrafish (*Danio rerio*) were maintained at Wayne State University School of Medicine. Fish were fed a combination of brine shrimp and dry food three times daily and maintained at 28.5°C under a light cycle of 14 hr light: 10 hr dark. Goldfish (*Carassius auratus*) were purchased from a local pet store. All protocols used in this study were approved by the Institutional Animal Care and Use Committee at Wayne State University School of Medicine (approval numbers: A12-05-12 and A03-02-13) and were performed in strict compliance with Institutional and NIH Guidelines.

## Phylogenetic and conserved synteny analysis

Phylogenetic analyses used EnsemblCompara GeneTrees (*Vilella et al., 2009*; *Flicek et al., 2013*). This method starts with data in Ensembl genomes, and uses the longest protein translation for each gene, identifies the Best Reciprocal BlastP Hit, clusters data into gene families, aligns sequences by MUSCLE (*Edgar, 2004*) and builds trees with TreeBeST (https://github.com/Ensembl/treebest), taking into account the multifurcated species tree, and the timing of gene duplication events judging from the distribution of each paralog/ortholog in species before and after each duplication event (*Vilella et al., 2009*). The tree we used is available at Ensembl under accession ENSGT00500000044795. The tree in *Figure 1* presents data after the collapse of large clades while retaining rayfin fish genes shown in detail. Conserved Synteny analysis used the Synteny Database (*Catchen et al., 2011a*, *2011b*).

## Reverse transcription polymerase chain reaction (RT-PCR) and quantitative RT-PCR

RNA was isolated from wild-type *D. rerio* embryos at multiple stages in development using the TRIZOL reagent (Ambion, Carlsbad, CA). Complementary DNAs were made using Oligo(dT) (Invitrogen) as a primer and Superscript II (Invitrogen) as a reverse transcriptase. The following primers were designed to span across an intron and amplify a 100–400 base pair product:

*plin6* (S: F'- ACG CGC TCT CAC GAA CGA CC −3'; R: 5'-CAG AGG CAC GGC CAG AGG GA −3').

Expression profiling of adult skin and adipose tissue was performed by quantitative RT-PCR in biological and technical triplicate, as previously described (*Thummel et al., 2010*). For *plin6* amplification, a PrimePCR Custom Assay (Biorad) was used for primer design as follows: *plin6* (F: 5' − CA TGGAGATGGTGGAGATGC −3'; R: 5'- CTCCTCCACATGATGAACCC −3'). Expression was normalized to that of *gpia* mRNA, using the custom primers in the PrimePCR SYBR Green Assay for *gpia* (Biorad; Assay ID: qDreCED0007433). Analysis was performed using the Livak $\Delta\Delta C(t)$ method (*Livak and Schmittgen, 2001*).

A partial *plin6* cDNA from goldfish was isolated from isolated skin scale mRNA by traditional RT-PCR using the following primer sequences that are conserved between zebrafish and pufferfish. Two sense primers (5'-cgg tgg gtc taa aga ggc tga tgg aga tg-3' and 5'- CTC TTC CTC AAG GCC ATG GA-3') were paired with a single antisense primer (5'- GCT CAG ATA CTC AAA CGC TCG CTG GCT CTG-3'). PCR products were cloned, sequenced and submitted to GenBank (accession # BankIt1599300 seq1 KC514068).

## Immunohistochemistry

Immunohistochemistry was performed as previously described (*Thummel et al., 2010*). Primary antibodies to zebrafish Plin6 were produced against the full-length recombinant protein, produced in bacteria, and affinity purified by a commercial service (Proteintech, Chicago, IL). The following primary antibodies and dilutions were used: rabbit anti-Plin6 antisera (1:1000) and mouse monoclonal anti-green fluorescent protein (GFP) antibody (1:500, Sigma Chemical, St. Louis, MO). Embryos were fixed overnight at 4°C in 4% paraformaldehyde (in 5% sucrose/1X PBS), washed in 5% sucrose/1X PBS three times for 20 min, incubated in 30% sucrose overnight at 4°C, then in a 1:2 dilution of 30% sucrose/Tissue Freezing Medium (TFM, Triangle Biomedical Sciences, Durham, NC) overnight at 4°C.

Following cryosectioning (14 μm), the sections were dried for 2 hr at 50°C, followed by rehydration in 1X PBS. Tissue sections were incubated in blocking solution (1X PBS/2% normal goat serum/1% DMSO/0.2% Triton-X 100) for 1 hr at room temperature and then in blocking solution containing the primary antibody overnight at 4°C. Next, the sections were washed in 1X PBS/0.05% Tween-20, incubated for 1 hr at room temperature in an Alexa Fluor-488, or −594 goat anti-primary secondary antisera (Invitrogen, Carlsbad, CA) diluted 1:500 in 1X PBS/0.05% Tween-20. Nuclei were labeled with TO-PRO-3 (Invitrogen). Analysis for all immunohistochemistry was performed using a Leica TCS SP2 confocal microscope or an Olympus IX81 microscope equipped with a spinning disc confocal unit.

## Western blot and phosphoproteomic analysis

Immunoblot analysis was performed as we have previously described (*Thomas et al., 2011*). Total protein was isolated from control and *plin6* mutants. The protein equivalent of one embryo was subjected to SDS PAGE and transferred to a PVDF membrane (Amersham). Blots were probed with either anti-Plin6 antisera (diluted 1:2000), HRP-conjugated anti-$\beta$-actin monoclonal antibody (diluted 1:10,000, Calbiochem), or HRP-conjugated GAPDH monoclonal antibody (diluted 1:5000, proteintech) overnight at 4°C in blocking buffer. Secondary antibody (HRP-conjugated goat anti-rabbit IgG, 1:5000) was detected by chemiluminescence with a FluorChem E gel and Western blot imaging system (Cell Biosciences). In a separate experiment, total protein was collected from adult zebrafish skin and purified CD (see below). 15 μg of protein from each sample was subjected to SDS PAGE and probed as described above. Phosphoproteomic analysis was performed using Acrylamide-pendent Phos-tag (Wako Chemicals; Richmond, VA), per the manufacturer's instructions. Total protein was isolated from zebrafish skin following treatment with 10 μM epinephrine (Sigma Chemical, St. Louis, MO) or 10 μM forskolin and 100 μM isobutylmethylxanthine (FSK-IBMX; Sigma Chemical). Proteins (15 μg) were resolved using polyacrylamide gels containing Phos-tag (100 μm) and $MnCl_2$ (200 μm), which retards the electrophoretic mobility of phosphorylated proteins. Secondary antibodies and detection was performed as described above.

## Carotenoid droplet purification

Scales and skin were homogenized in 20 mM HEPES buffer (pH 7.5) containing 40% sucrose and protease inhibitors. Total homogenate (1 ml) was overlaid with 1.1 ml of buffer containing 20% sucrose, and centrifuged at 100,000 x g for 30 min. Protein fractions were recovered at the interface of the 20% and 40% sucrose layers, and the orange layer containing CD that floated above 20% sucrose. The purified CD fraction was precipitated with acetone, and proteins subjected to standard immunoblot analysis as described.

## Analysis of CD and Plin6 trafficking in xanthophores

Live cell imaging was performed as previously described for mammalian cells (*Granneman et al., 2007*, *2009*). Briefly, isolated scales from adult zebrafish were maintained in basal medium or pretreated for 1 hr with epinephrine to fully aggregate carotenoid droplets. Scales were then placed in a recording chamber containing fresh M199 media without phenol red and imaged at room temperature using an automated Olympus IX81 microscope equipped with a spinning disc confocal unit. Brightfield and fluorescence images were captured using an Olympus 40 × 0.9 NA apochromatic water immersion lens and a Hamamatsu ORCA cooled CCD camera. After image collection, scales in the experimental group were treated with 20 μM forskolin and 200 μM isobutylmethylxanthine (FSK-IBMX) and re-imaged 1 hr post-stimulation. In other experiments, scales were treated, post-fixed, and processed for immunofluorescence using Plin6 antisera, as described above. The Pigmentary Organelle Index (*Taylor et al., 1991*) was used for semi-quantitative analysis of CD trafficking by autofluorescence (FITC filter set) and Plin6 immunofluoresce signals. Scoring of CD trafficking was performed by an analyst who was blinded to the treatment conditions.

## Thin section transmission electron microscopy (TSTEM)

Zebrafish xanthophores were isolated and prepared as described for goldfish (*Kimler et al., 1993*) with minor modifications. Briefly, scales were collected and were treated with epinephrine (10 μM) or FSK-IBMX to aggregate or disperse xanthophore CD, respectively. Scales were fixed in 4% paraformaldehyde with light glutaraldehyde (0.05%) in PHEM buffer (pH 7.4) (*Kimler et al., 1993*), *en*

*bloc*-stained in 30% ethanol, 2% uranyl acetate, and dehydrated in a graded series of ethanol. Scales were transitioned into LR-White (medium hardness) embedding resin (Sigma Chemical) in Beem capsules (Ted Pella, Redding, CA) and incubated in a dry oven (70°C) overnight to polymerize the embedding medium. Thin sections (70–90 nm) were cut on a RMC 6000 ultramicrotome (Boeckeler Instruments, Tucson, AZ) and placed on Formvar-coated gold finder grids. Sections were immunolabeled with rabbit anti-Plin6 antisera (3 µg/ml). Goat anti-rabbit IgG Fluoronanogold™ (FNG) probes (1.4 nm; 1/25 dilution; Nanoprobes, Yaphank, NY) were used for secondary antibody labeling. Labeled xanthophores were examined by confocal microscopy and grid coordinates were noted for correlative thin-section TEM (*Granneman et al., 2011*). After secondary fixation in 3% glutaraldehyde with 4% tannic acid in 0.12M sodium cacodylate buffer (pH 7.5) FNG probes were silver-enhanced to 10–15 nm and gold toned. Cells were post-fixed with osmium tetroxide and post-stained with uranyl acetate and Reynolds lead citrate. Imaging was performed on a JEOL-2010 Fas-TEM at 200kV.

## Generation of *plin6* mutant zebrafish

A TALEN was designed and synthesized by Transposagen Biopharmaceuticals, Inc. (Lexington, KY) to target an *Acc*I restriction site on exon 3 of *plin6*. The sequence is as follows: 5' – TGCGCTGCAG-CAGGCCTC<u>ATCCGTCTACACAGTT</u>GTAAAAGGACGGTATCCA – 3' (target sequence underlined). 5 nl of the TALEN mRNA (100 pg/nl) was micro-injected into 1-cell-stage embryos and NHEJ repair events were confirmed the following day in a subset of injected embryos by PCR/restriction analysis. Cohorts of successful injections were raised for 1 month, when individuals were genotyped from tail snips, and the disrupting mutation confirmed by DNA sequencing. Heterozygous mutants were raised to sexual maturity and bred to wild-type fish to verify germline transmission, then appropriate matings were performed to obtain null and wild-type fish for analysis.

## Analysis of scale and skin carotenoids and triglyceride levels

Zebrafish skin was collected in biological quadruplicate for wild-type and mutant animals, washed twice with cold PBS and centrifuged. Tissue was then transferred to a 10 ml glass tube and extracted twice with 5 ml of acetone. Extracts were collected after centrifugation, dried under nitrogen stream, and resuspended in 0.5 ml of petroleum ether. Extracted carotenoids were quantified by comparing absorbance values at 430 nm to known beta carotene standard. One representative sample of wild-type and mutant absorbance is shown in the corresponding figure.

Integumentary triacylglycerol (TAG) levels were determined by pulverizing washed skin and scales of individual fish (N = 3) in liquid nitrogen, then extracting total lipids with chloroform/methanol, as previously described (*Bligh and Dyer, 1959*). Extracts were dried under nitrogen, and suspended in 200 µl of 5% NP40. TAG of the extracts was solubilized by twice heating fractions to 100°C and then cooling to room temperature. Extracts were centrifuged at 1000 x g, diluted 10-fold in water, and assayed for TAG content using a commercial kit Triglyceride Determination Kit (Sigma, Cat.TR0100) according to the manufacturer's protocol.

## Generation of fluorescent fusion proteins and LD formation in COS-7 cells

Zebrafish *plin6* was amplified by PCR and cloned into the mammalian expression vector EYFP-C1 (Clontech). The mouse PLIN1-ECFP expression vector was previously described (*Granneman et al., 2011*). COS-7 cells were purchased from American Type Culture Collection (ATCC) and were validated by the manufacturer using standard techniques. COS-7 cells were transfected with zebrafish EYFP-Plin6 alone or zebrafish EYFP-Plin6 with mouse PLIN1-ECFP at a plasmid ratio of 1:1. Following transfection, cells were incubated with media supplemented with 400 µM oleic acid complexed to BSA for 18 hr as previously described (*Granneman et al., 2011*). Neutral lipids were stained with LipidTOX (Invitrogen, 1:200). Cells were fixed in 1% paraformaldehyde, transferred onto coverslips, and imaged with an Olympus IX81 microscope, as described above.

## Acknowledgements

Funding sources for this work include: National Institutes of Health grants RO1DK076629 (JG), RO1DK062292 (JG), R21EY019401 (RT), P30EY04068 (RT), 5R01OD011116 (JHP), an unrestricted

grant from Research to Prevent Blindness to Wayne State University, Department of Ophthalmology (RT), and a GrantsPlus enhancement grant (JG). The funding sources had no role in study design, data collection and analysis, decision to publish, or preparation of the manuscript.

We thank Li Zhou for technical assistance and Drs. John D Taylor, TT Tchen, and Andrew S Greenberg for insights into the similarities between LD and CD.

## Additional information

### Funding

| Funder | Grant reference number | Author |
| --- | --- | --- |
| National Institute of Diabetes and Digestive and Kidney Diseases | RO1DK076629 | James G Granneman |
| Wayne State University | Grants Plus,Start-up funds | James G Granneman Ryan Thummel |
| National Institute of Diabetes and Digestive and Kidney Diseases | RO1DK62292 | James G Granneman |
| NIH Office of the Director | 5R01OD011116 | John H Postlethwait |
| National Eye Institute | R21EY019401 | Ryan Thummel |
| Research to Prevent Blindness | Unrestricted Grant | Ryan Thummel |
| National Eye Institute | P30EY04068 | Ryan Thummel |

The funders had no role in study design, data collection and interpretation, or the decision to submit the work for publication.

### Author contributions

JGG, Conceptualization, Resources, Software, Formal analysis, Supervision, Validation, Investigation, Methodology, Writing—original draft, Project administration, Writing—review and editing; VAK, Conceptualization, Investigation, Methodology, Formal analysis; HZ, XY, XL, Investigation, Methodology; JHP, Resources, Software, Investigation, Methodology, Writing—original draft, Writing—review and editing; RT, Conceptualization, Resources, Formal analysis, Supervision, Validation, Investigation, Methodology, Writing—original draft, Project administration, Writing—review and editing

### Author ORCIDs

Ryan Thummel, http://orcid.org/0000-0002-0522-8704

### Ethics

Animal experimentation: All protocols used in this study were approved by the Institutional Animal Care and Use Committee at Wayne State University School of Medicine (approval numbers: A12-05-12 and A03-02-13) and were performed in strict compliance with Institutional and NIH Guidelines.

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
