## [Decision Letter]

Thank you for submitting your article "Lipid Droplet Biology and Evolution Illuminated by the Characterization of a Novel Perilipin in Teleost Fish" for consideration by *eLife*. Your article has been favorably evaluated by Vivek Malhotra (Senior Editor) and three reviewers, one of whom is a member of our Board of Reviewing Editors. The following individual involved in review of your submission has agreed to reveal their identity: Axel Meyer (Reviewer #3).

The reviewers have discussed the reviews with one another and the Reviewing Editor has drafted this decision to help you prepare a revised submission.

Summary:

The authors identified a new perilipin protein that is found only in teleosts but that has homologs in all vertebrates. These proteins are involved in lipid droplet formation and triglyceride metabolism. The human genome contains at least 5 genes coding for these proteins. The teleost specific perilipin, called *plin6*, arose during the whole genome duplications that occurred in the early phases of vertebrate evolution. Plin6 is expressed in specialized skin cells that are involved in red/yellow pigmentation and not in cells where they would be expected to play a role, for instance in cells involved in lipid metabolism where the other members of this family are usually expressed. The authors report data showing that *plin6* is regulated by protein kinase A (PKA), a kinase known to regulate lipid droplets. Other experiments show that the deletion of *plin6* impairs the ability of pigment (carotenoid) droplets to cluster. The authors provide structural evidence, in addition to the role of perilipin, that lipid droplets maybe be homologous to pigment droplets.

The reviewers have provided very positive and constructive comments that are required to clarify the result presentation and to better support some of the reported findings.

Essential revisions:

1) The presentation of the phylogenetic analyses should be improved. The results are rather difficult to follow in some sections. For instance, in the first paragraph the subsection “Teleost fish contain a unique PLIN variant, Plin6, which arose from the first two vertebrate genome duplications” it is implied that several PLIN genes originated during the two WGD events. However, a certain number of genes were there before. Which ones come from the WGD and have been maintained as ohnologs and which ones were preserved a single copy and in which lineage? This needs to be clarified, maybe with a simplified figure, especially for *plin6*. Was *plin6* the ohnolog of another gene and was it only maintained in some fish lineages?

2) Subsection “Teleost fish contain a unique PLIN variant, Plin6, which arose from the first two vertebrate genome duplications”, second paragraph: some technical terms are used and they would need to be defined or simplified. For instance, "Comparatree analysis". Is this a software, a phylogenetic method? I would expect more details to be provided in the Materials and methods because the reader needs to understand what is to be interpreted from the results provided. Resolving the history of gene duplication, gains and losses etc. is not an easy task and not enough details are provided to actually support the results.

3) Plin6 does appear to be expressed in developing zebrafish larvae and in adult skin. However, it would be important to have more detailed data about the expression pattern. In which tissues is *plin6* expressed? Here, in situ hybridization or wholemount staining at larval stages and/or qPCR for a few tissues (including skin, muscle and adipose tissue) would be important data to show. Since i.e. Plin 5 and 1 seem to be mainly restricted to muscles and fat cells respectively, comparing the expression levels of all Plins in muscle, adipose tissue and skin would provide further evidence that Plin6 function has diverged from Plin1 and 5. This additional data would be helpful, since only so little is known about Plin proteins outside of mammals.

4) The use of *fit2* in Figure 3 should be clarified to demonstrate relative expression of *plin6*, which is implicated in fat storage and seems to have its highest expression in adipose tissue. Why was a more common control gene such as b-actin (or a few housekeeping genes) not used?

5) The authors report that there are no apparent systemic defects in mutant embryos, larvae, or adults of *plin6*. It would be useful to see pictures of larvae and adult fish also to see the difference in carotenoid concentration in fins and other parts of the body.

---

## [Author Response]

*Essential revisions:*

*1) The presentation of the phylogenetic analyses should be improved. The results are rather difficult to follow in some sections. For instance, in the first paragraph the subsection “Teleost fish contain a unique PLIN variant, Plin6, which arose from the first two vertebrate genome duplications” it is implied that several PLIN genes originated during the two WGD events. However, a certain number of genes were there before. Which ones come from the WGD and have been maintained as ohnologs and which ones were preserved a single copy and in which lineage? This needs to be clarified, maybe with a simplified figure, especially for plin6. Was plin6 the ohnolog of another gene and was it only maintained in some fish lineages?*

We have extensively edited and revised this section of the manuscript. In addition, we have included a new figure to outline the gene history of the entire PLIN family. Please see the amended text for more detail. To directly answer the reviewers’ questions, all of the existing vertebrate *plin* genes come from a single gene in a chordate ancestor. Ciona has two genes today but those are likely from a gene duplication after the divergence of urochordates from vertebrates. The reviewer is correct, *plin6* is an ohnolog of *plin1, plin2*, and *plin345*; it was only maintained in the ray fin lineage. Again, we have clarified the text to all of these points.

*2) Subsection “Teleost fish contain a unique PLIN variant, Plin6, which arose from the first two vertebrate genome duplications”, second paragraph: some technical terms are used and they would need to be defined or simplified. For instance, "Comparatree analysis". Is this a software, a phylogenetic method? I would expect more details to be provided in the Materials and methods because the reader needs to understand what is to be interpreted from the results provided. Resolving the history of gene duplication, gains and losses etc. is not an easy task and not enough details are provided to actually support the results.*

While the original text provided a reference that describes how Compara GeneTree analysis works, we recognize that the reviewer simply points out that other readers will have the same issue; thus, the revision contains a quick synopsis in the Materials and methods. In addition, we define carefully other terms, such as ‘ohnologs’.

*3) Plin6 does appear to be expressed in developing zebrafish larvae and in adult skin. However, it would be important to have more detailed data about the expression pattern. In which tissues is plin6 expressed? Here, in situ hybridization or wholemount staining at larval stages and/or qPCR for a few tissues (including skin, muscle and adipose tissue) would be important data to show. Since i.e. Plin 5 and 1 seem to be mainly restricted to muscles and fat cells respectively, comparing the expression levels of all Plins in muscle, adipose tissue and skin would provide further evidence that Plin6 function has diverged from Plin1 and 5. This additional data would be helpful, since only so little is known about Plin proteins outside of mammals.*

We have added a new figure to address this concern (see new Figure 3—figure supplement 1). Specifically, we provide a new Western blot of Plin6 in skin, muscle, liver, and gut tissue to demonstrate that Plin6 expression is restricted to skin and is not present in the other tissues. While we agree that little is known about the other Plin proteins outside of mammals, we felt that initiating the analysis of all four Plin proteins was outside the realm of this manuscript, so we focused on providing stronger evidence that Plin6 is restricted to skin.

*4) The use of fit2 in Figure 3 should be clarified to demonstrate relative expression of plin6, which is implicated in fat storage and seems to have its highest expression in adipose tissue. Why was a more common control gene such as b-actin (or a few housekeeping genes) not used?*

Thank you, and we agree. We have repeated the qPCR shown in Figure 3 using the housekeeping gene *gpia* as a more appropriate control.

*5) The authors report that there are no apparent systemic defects in mutant embryos, larvae, or adults of plin6. It would be useful to see pictures of larvae and adult fish also to see the difference in carotenoid concentration in fins and other parts of the body.*

We have added a new figure to address this concern (see new Figure 6—figure supplement 2) that provided pictures of wild-type and *plin6* mutant larvae and adult caudal fin.